# A Rare Case of Post-Traumatic Cervical Ligamentous Tear Complicated by Vertebral Arteriovenous Fistula (vAVF), with Successful Endovascular Treatment

**DOI:** 10.3390/diagnostics13162693

**Published:** 2023-08-16

**Authors:** Abdul Rahim Nur Fazdlin, Iqbal Hussain Rizuana, Li Shyan Ch’ng

**Affiliations:** 1Department of Radiology, Faculty of Medicine, University Kebangsaan, Bangi 43600, Malaysia; 2Department of Radiology, Hospital Kuala Lumpur, Kuala Lumpur 50586, Malaysia; 3Hospital Canselor Tuanku Muhriz, Jalan Yaacob Latif, Bandar Tun Razak, Kuala Lumpur 56000, Malaysia; 4Department of Radiology, Faculty of Medicine, Universiti Teknologi Mara, Sungai Buloh 47000, Malaysia; lishyanc@yahoo.com

**Keywords:** vertebral arteriovenous fistula, traumatic, endovascular

## Abstract

Post-traumatic vertebral arteriovenous fistula (vAVF) caused by motor vehicle accidents (MVA) is a rare condition in which there is abnormal communication between the vertebral artery and its adjacent veins. In a post-MVA setting, it is commonly associated with vertebral body fracture. In this paper, we report a case of a 19-year-old girl with a complete C2/C3 anterior and posterior ligament tear post MVA without any cervical bony injury. Initial plain computed tomography (CT) cervical scan showed a prevertebral hematoma. A CT angiogram (CTA) raised the suspicion of a pseudo-aneurysm at the right posterior C3 vertebral body. Further imaging with magnetic resonance imaging (MRI) demonstrated traumatic AVF at the C2/C3 level involving the V2/V3 right vertebral artery to the vertebral venous plexus. Digital Subtraction Angiography (DSA) further revealed a transected right vertebral artery at the C2/C3 level with an arteriovenous fistula and an enlarged vertebral venous plexus. The fistulous communication was successfully occluded with coils from a cranial and caudal approach to the transected segment right vertebral artery, with a total of eight coils. Post-MVA vertebral arteriovenous fistula (vAVF) is a rare sequela of vertebral bony injury at the cervical region, and is an even rarer association with an isolated ligamentous injury, whereby endovascular treatment with ipsilateral vertebral artery closure is a feasible treatment of vAVF.

Vertebral arteriovenous fistula (vAVF) post-motor vehicle accident (MVA) is a rare condition caused by abnormal communication between the vertebral artery with the adjacent veins [1]. It can be spontaneous in origin when it is due to underlying vascular pathologies related to connective tissue disorders (e.g., Neurofibromatosis, Ehler–Danlos disease, Marfan’s syndrome), or it can be traumatic in origin [2]. Spinal Dural Arteriovenous Fistula (SDAVF) is a rare incidence in a trauma setting, and it commonly occurs at the thoracolumbar level [3,4]. The fistula is typically found in the dural sleeve of an exiting nerve root that receives its arterial blood from the meningeal branch of a segmental artery, and subsequently drains into the coronal venous plexus [4,5]. We report a unique case of post-MVA vAVF following an anterior and posterior cervical ligament tear with no associated bony injury, which was subsequently treated successfully with endovascular coiling.

A 19-year-old girl, with an alleged MVA, presented with severe traumatic brain injury, and she was subsequently intubated. Plain computed tomography (CT) cervical showed prevertebral cervical hematoma without any cervical bony injury. A CT angiogram (CTA) of the neck and the thorax showed a suspicious pseudoaneurysm at the right posterior C3 vertebral body level, causing mass effect on the spinal cord. Magnetic resonance imaging (MRI) showed a complete anterior and posterior ligamentous tear at the C2/3 level with traumatic AVF involving the V2/V3 right vertebral artery to the vertebral venous plexus (Figure 1A–C). The patient then underwent Digital Subtraction Angiography (DSA), which revealed a transected right vertebral artery at the C2/C3 level, with an arteriovenous fistula and an enlarged vertebral venous plexus. The fistulous communication was successfully occluded with coils from a cranial and caudal approach. For the occlusion, eight Cosmos Microplex coils (Microvention Inc., Aliso Viejo, CA, USA) were used (four 6 mm × 26 cm coils, three 7 mm × 31 cm, and one 8 mm/37 cm coils), and they were deployed via an Echelon-10 microcatheter (eV3, Medtronic, MN, USA) within the fistula and the right vertebral artery (Figure 1D–F).

Spinal dural arteriovenous fistula (sDAVF), a rare condition, which is shown to have a male predominance, accounts for 60–80% of all spinal vascular malformations [4,6]. Clinically, they present with myelo- and radiculopathies, which are sometimes misdiagnosed as degenerative disc disease, myelitis, or an intramedullary tumor [4,6]. There are two types of sDAVF––traumatic and spontaneous. Spontaneous causes are present in sDAVF due to congenital weakness/dysplasia of the wall, whereas traumatic causes are a result of blunt penetrating or iatrogenic trauma.

With the presence of fistulous communication, increased pressure within the arterialized medullary veins results in venous hypertension, and it subsequently causes venous engorgement from caudal to cranial. Venous engorgement causes a pressure effect on the spinal cord, leading to cord oedema and a risk of ischaemia/infarction [4,6]. Lower cervical fistulas may show a localizing sign of upper extremity weakness or cranial nerve involvement in a conscious patient [4]. Other presenting symptoms include the presence of subarachnoid hemorrhages, dizziness, or a progression of one of the initial symptoms. These happen because there is a decline in spinal function following aggravation of the cord oedema from venous congestion [4]. While spontaneous AVFs typically include the third segment, where the artery exits the foramen of the atlas and enters the foramen magnum, the majority of traumatic vertebral AVFs involve the long second portion (intraforaminal) of the vertebral artery [7].

In intubated patients, such as in our case, it is important to recognize initial imaging findings to enable early intervention and improve morbidity. Traditionally, prone–supine myelography has been the standard form of examination, which would show enlarged serpentine-filling defects representing a dilated venous plexus with or without tortuosity [5]. These prominent vessels are better seen on a tomogram at three consecutive levels, and they extend over an average of seven to eight levels. Lumbar sDAVF may show beading of the cauda equina, with the majority of cases showing involvement of the conus medullaris tip [5]. On MRI, the most sensitive finding is a homogenous hyperintense T2 intramedullary signal centrally representing centromedullary oedema with associated cord expansion [5,6], and this was seen in our patient and extended across three levels (from the C1/2 to the C3/C4 levels).

On T2-weighted images, there will be a serpiginous dorsal subarachnoid flow void signal representing dilated perimedullary vessels, which is seen in 45% of the cases [5,6]. Our patient demonstrated an enlarged spinal epidural venous plexus from the C1/2 to the C3/4 levels. There was associated thickening of the adjacent V3 vessel wall in keeping with an intramural hematoma. These findings suggest that the injury likely originated at the junction of the V2/3 segment of the vertebral artery.

A few studies have shown that prominent flow voids on T2-weighted images at the dorsal surface of the cord are among the first initial findings of sDAVF. This flow void signal will demonstrate a scalloped and irregular cord appearance on T1-weighted images [5].

Mass effect is an uncommon presentation as it is shown in less than 50% of cases. Abnormal enhancement can be seen within the dilated venous plexus and the intramedullary, and in post-contrast study [5]. However, although uncommon, the diagnosis of sDAVF should also be considered in the presence of T2 intramedullary hyperintense signal, flow voids, and venous plexus enhancement with mass effect [5]. In our case, mass effect was seen together with the flow voids, which was thought to be the presence of a pseudoaneurysm at the C2/3 levels, hence the fact that we proceeded with an angiography study.

Gadolinium is vital in MR spine examination with myelopathy. Studies have shown that its usage helps to enhance the sensitivity and the specificity of the MRI up to 88% positivity among patients with SDAVF [5].

The ‘gold standard’ in diagnosing sDAVF is spinal angiography [4]. Unfortunately, the procedure is not always simple because low-flow-volume fistulas may be difficult to detect and are often missed. Another potential complication is the worsening of venous congestion, causing rapid clinical deterioration due to angiography, which leads to emergency surgery [4].

Prior to the angiography study, evaluation with MR and CTA is mandatory to correctly identify the fistula, especially if it is a slow-flow fistula. However, three-dimensional (3D) rotational angiography equipment would be more effective in identifying a complex spinal vascular architecture. The primary treatment goal is to occlude the fistula, preventing communication between the artery and the vein while preserving the vertebral artery [2].

In our patient, the fistulous communication was correctly identified and successfully occluded with a total of eight detachable coils deployed within the fistula and the right vertebral artery via a cranial to caudal approach. The coils were also deployed caudally via the left vertebral artery to ensure the pseudoaneurysm would not be filled retrogradely via the left vertebral artery into the cranial portion of the right vertebral artery. Various sizes of coils were used to achieve adequate occlusion and packing, thus preventing recanalization. Different coil sizes were needed for framing and packing in order to prevent coil migration. A post-coiling angiogram showed complete occlusion of the fistula with non-opacification of the dilated and tortuous venous plexus.

In addition to coiling, endovascular balloon occlusion is a feasible endovascular treatment option, as described by Briganti et al. However, recurrence of fistula may occur due to the deflation of the balloon. Stent graft in the treatment of sDAVF would preserve vessel patency, but it might not be feasible in very rigid and tortuous vessels [8]. Other disadvantages of a covered stent include in-stent thrombosis and incomplete closure of the fistula due to malposition [9]. Particulate embolic agents and cyanoacrylate glue are not favored in high-flow fistulas due to the risk of distal migration and infarct [10]. In addition to the transarterial approach, transvenous embolization could be attempted to preserve the vertebral artery [11]. Endovascular treatment for sDAVF is favored over surgery due to complications post-surgery, such as steal flow symptoms, massive hemorrhages, and injuries to surrounding structures [12].

Prior to occlusion, a left vertebral angiogram was performed to assess posterior circulation and feasibility to sacrifice the right vertebral artery during the intervention procedure. In this case, the left vertebral artery showed good supply to both sides of the posterior fossa and to the V4 segment of the right vertebral artery. Flow to the V4 segment of the right vertebral artery is essential to prevent right posterior inferior cerebellar territory infarction.

After treatment, we were unable to assess for clinical improvement in our patient as the patient was intubated due to severe traumatic brain injury as a sequela of the MVA. According to various studies in the literature, significant clinical improvement has been achieved amongst patients who presented acutely and had timely management [6]. Patients who showed poor clinical outcome during follow-up were those with delayed management. Our patient had a modified Rankin (mRankin) score of 5, as there were multiple intracranial hemorrhages resulting from the trauma, which contributed to her poor outcome.

In conclusion, post-MVA vertebral arteriovenous fistula (vAVF) is a rare sequela of vertebral bony injury at the cervical region and an even rarer sequelae of an isolated ligamentous injury. Imaging is essential, especially among intubated patients, because early detection is vital to aiding further management and minimizing complications. Few works in the literature support the notion that MR is the current first-line imaging modality in diagnosing sDAVF. Findings of hyperintense T2W signal with enlarged and enhanced coronal venous plexus provide a high probability of vascular malformation. Hence, spinal angiography would be the next step of management [5]. Endovascular treatment with ipsilateral vertebral artery occlusion is a feasible treatment for vAVF. 

## Figures and Tables

**Figure 1 diagnostics-13-02693-f001:**
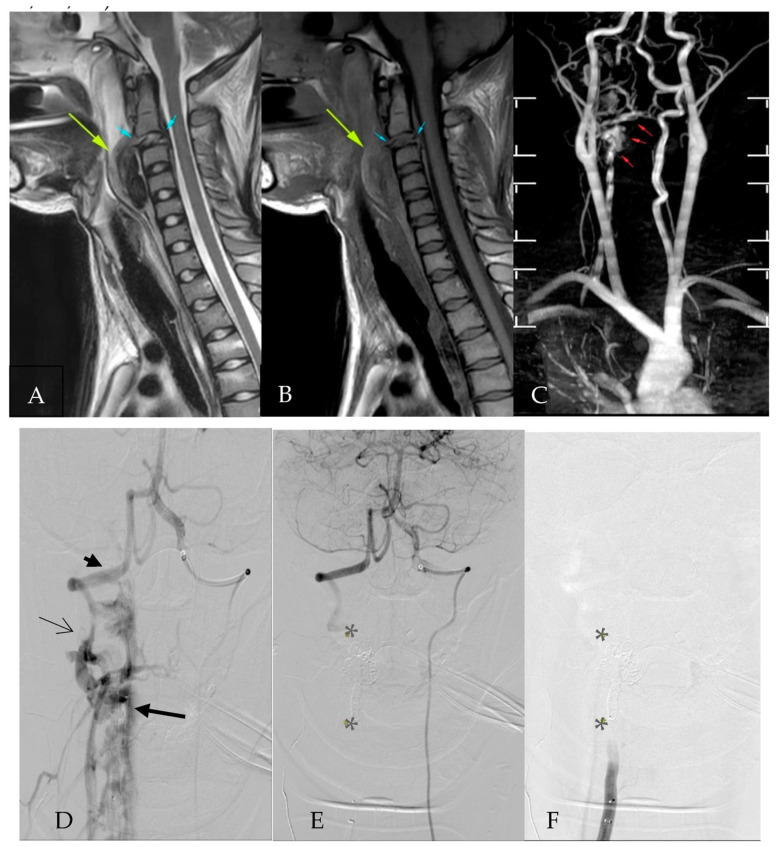
(**A**,**B**): MRI T1W/T2W showing a complete tear of the C2/3 anterior and the posterior ligaments (blue arrow) with a prevertebral soft tissue hematoma (yellow arrow) at the C2–C4 levels. (**C**): MRA reconstruction showing early opacification of the right vertebral venous plexus (red arrow) representing a vertebral arteriovenous fistula (vAVF). (**D**): Left vertebral pre-coiling angiogram showing reflux into the right vertebral artery (arrowhead) and fistulous communication (thin arrow) with the right vertebral plexus (thick arrow). (**E**): Post-coiling angiogram showing non-opacification of the right vertebral venous plexus with coils in situ (*). (**F**): Right vertebral angiogram showing no forward flow of right vertebral artery and areas of coils (*).

## Data Availability

Not applicable.

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
