# Peer review of "A Rare Case of Post-Traumatic Cervical Ligamentous Tear Complicated by Vertebral Arteriovenous Fistula (vAVF), with Successful Endovascular Treatment"

_diagnostics, 2023, doi:10.3390/diagnostics13162693_

Round 1
Reviewer 1 Report
A competent case report with high clinical interest. No structural or contextual defects noticed.
Some specific comments below:
1. No main question addressed. it is a case report.
2. The case report is not an "original idea", but has a high relevance to the field
3. No specific gap in the field can be addressed by a case report.
4. It is a rare disease case report which provides images and clinical evidence necessary for field communication.
5. The references are appropriate.
6. Figures are adequate to support the main message.
Author Response
Some specific comments below:
- No main question addressed. it is a case report.
- The case report is not an "original idea", but has a high relevance to the field.
- No specific gap in the field can be addressed by a case report.
- It is a rare disease case report which provides images and clinical evidence necessary for field communication.
- The references are appropriate.
- Figures are adequate to support the main message.
Response : Thank you.
Reviewer 2 Report
This Case Report lacks the standard methodology sections used. Breaking down the information into distinct subsections can aid readability. This could include sections such as "Introduction", "Case Presentation", "Discussion", "Treatment", and "Conclusion".
Abstract
1. There are several repetitions of the term "vertebral arteriovenous fistula (vAVF)" and "post MVA". Once you have defined the abbreviation (vAVF), you can use it throughout without repeating the full term.
2. It's typical to introduce an acronym the first time a term is used, then use the acronym consistently throughout. You've done this for vAVF and MVA, but not for CT, CTA, MRI, and DSA.
3. The abstract mentions a "pseudo aneurysm" without explaining what it is. If it's a crucial part of your case, it should be clarified
4. There's a strange sentence in "The fistulous communication was successfully occluded with coils from cranial and caudal approach to the transected segment right vertebral artery with a total of 8 coils." This sentence is confusing and could be broken down into simpler, shorter sentences for better readability.
Manuscript
1. There are a few instances of unnecessary punctuation, such as the period after the citation in brackets. The period should come after the brackets, not before.
2. Once you've defined an acronym, use it consistently throughout the text. For example, after defining spinal dural arteriovenous fistula as "sDAVF", you should use the acronym.
3. Be careful with singular and plural usage. For instance, "8 Cosmos Microplex coils (Microvention Inc, USA) was used" should be "were used".
4. "3-Dimensional (3D) rotational angiography" is introduced in the text. Make sure it's abbreviated as "3D" throughout the text afterwards.
5. English - The past tense is generally used for describing specific research or observations. In a few instances, the present tense is used where past tense would be more appropriate.
The past tense is generally used for describing specific research or observations. In a few instances, the present tense is used where past tense would be more appropriate.
Author Response
Please see the attached file with point-by-point responses.

Reviewer 3 Report
Very nice case presentation of an uncommon entity but with major implications.
Author Response
Comment : Very nice case presentation of an uncommon entity but with major implications.
Response : Thank you for the support.
Round 2
Reviewer 2 Report
Authors have made the requested changes. The paper can now be published